# Uncertainty Estimation and Calibration with Finite-State Probabilistic RNNs

**Cheng Wang**[*†], **Carolin Lawrence**[*], **Mathias Niepert**
NEC Laboratories Europe
{cheng.wang, carolin.lawrence, mathias.niepert}@neclab.eu

## Abstract

Uncertainty quantification is crucial for building reliable and trustable machine learning systems. We propose to estimate uncertainty in recurrent neural networks (RNNs) via stochastic discrete state transitions over recurrent timesteps. The uncertainty of the model can be quantified by running a prediction several times, each time sampling from the recurrent state transition distribution, leading to potentially different results if the model is uncertain. Alongside uncertainty quantification, our proposed method offers several advantages in different settings. The proposed method can (1) learn deterministic and probabilistic automata from data, (2) learn well-calibrated models on real-world classification tasks, (3) improve the performance of out-of-distribution detection, and (4) control the exploration-exploitation trade-off in reinforcement learning. An implementation is available.[1]

## 1 Introduction

Machine learning models are well-calibrated if the probability associated with the predicted class reflects its correctness likelihood relative to the ground truth. The output probabilities of modern neural networks are often poorly calibrated (Guo et al., 2017). For instance, typical neural networks with a softmax activation tend to assign high probabilities to out-of-distribution samples (Gal & Ghahramani, 2016b). Providing uncertainty estimates is important for model interpretability as it allows users to assess the extent to which they can trust a given prediction (Jiang et al., 2018). Moreover, well-calibrated output probabilities are crucial in several use cases. For instance, when monitoring medical time-series data (see Figure 1(a)), hospital staff should also be alerted when there is a low-confidence prediction concerning a patient's health status.

Bayesian neural networks (BNNs), which place a prior distribution on the model's parameters, are a popular approach to modeling uncertainty. BNNs often require more parameters, approximate inference, and depend crucially on the choice of prior (Gal, 2016; Lakshminarayanan et al., 2017). Applying dropout both during training and inference can be interpreted as a BNN and provides a more efficient method for uncertainty quantification (Gal & Ghahramani, 2016b). The dropout probability, however, needs to be tuned and, therefore, leads to a trade-off between predictive error and calibration error.

Sidestepping the challenges of Bayesian NNs, we propose an orthogonal approach to quantify the uncertainty in recurrent neural networks (RNNs). At each time step, based on the current hidden (and cell) state, the model computes a probability distribution over a finite set of states. The next state of the RNN is then drawn from this distribution. We use the Gumbel softmax trick (Gumbel, 1954; Kendall & Gal, 2017; Jang et al., 2017) to perform Monte-Carlo gradient estimation. Inspired by the effectiveness of temperature scaling (Guo et al., 2017) which is usually applied to trained models, we learn the temperature $\tau$ of the Gumbel softmax distribution during training to control the concentration of the state transition distribution. Learning $\tau$ as a parameter can be seen as entropy regularization (Szegedy et al., 2016; Pereyra et al., 2017; Jang et al., 2017). The resulting model, which we name ST-$\tau$, defines for every input sequence a probability distribution over state-

---

[*]Equal contribution.
[†]Work done at NEC Laboratories Europe.
[1]https://github.com/nec-research/st_tau

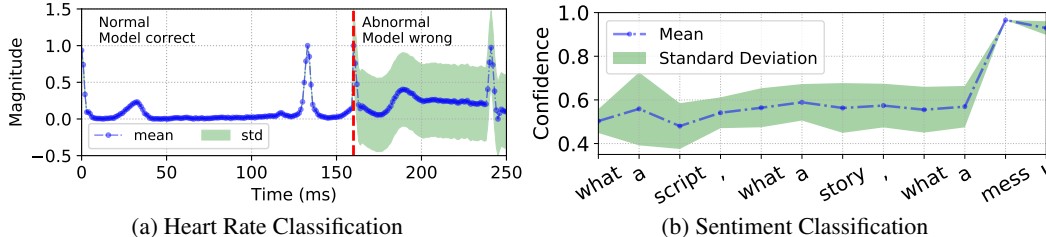

(a) Heart Rate Classification  (b) Sentiment Classification

Figure 1: (a) Prediction uncertainty of ST-$\tau$, our proposed method, for an ECG time-series based on 10 runs. To the left of the red line ST-$\tau$ classifies a heart beat as normal. To the right of the red line, ST-$\tau$ makes wrong predictions. Due to its drop in certainty, however, it can alert medical personnel. (b) Given a sentence with negative sentiment, ST-$\tau$ reads the sentence word by word. The y-axis presents the model's confidence of the sentence having a negative sentiment. After the first few words, the model leans towards a negative sentiment, but is uncertain about its prediction. After the word "*mess*," its uncertainty drops and it predicts the sentiment as negative.

transition paths similar to a probabilistic state machine. To estimate the model's uncertainty for a prediction, ST-$\tau$ is run multiple times to compute mean and variance of the prediction probabilities.

We explore the behavior of ST-$\tau$ in a variety of tasks and settings. First, we show that ST-$\tau$ can learn deterministic and probabilistic automata from data. Second, we demonstrate on real-world classification tasks that ST-$\tau$ learns well calibrated models. Third, we show that ST-$\tau$ is competitive in out-of-distribution detection tasks. Fourth, in a reinforcement learning task, we find that ST-$\tau$ is able to trade off exploration and exploitation behavior better than existing methods. Especially the out-of-distribution detection and reinforcement learning tasks are not amenable to post-hoc calibration approaches (Guo et al., 2017) and, therefore, require a method such as ours that is able to calibrate the probabilities during training.

## 2 UNCERTAINTY IN RECURRENT NEURAL NETWORKS

### 2.1 BACKGROUND

An RNN is a function $f$ defined through a neural network with parameters $\mathbf{w}$ that is applied over time steps: at time step $t$, it reuses the hidden state $\mathbf{h}_{t-1}$ of the previous time step and the current input $\mathbf{x}_t$ to compute a new state $\mathbf{h}_t$, $f : (\mathbf{h}_{t-1}, \mathbf{x}_t) \to \mathbf{h}_t$. Some RNN variants such as LSTMs have memory cells $\mathbf{c}_t$ and apply the function $f : (\mathbf{h}_{t-1}, \mathbf{c}_{t-1}, \mathbf{x}_t) \to \mathbf{h}_t$ at each step. A vanilla RNN maps two identical input sequences to the same state and it is therefore not possible to measure uncertainty of a prediction by running inference multiple times. Furthermore, it is known that passing $\mathbf{h}_t$ through a softmax transformation leads to overconfident predictions on out-of-distribution samples and poorly calibrated probabilities (Guo et al., 2017). In a Bayesian RNN the weight matrices $\mathbf{w}$ are drawn from a distribution and, therefore, the output is an average of an infinite number of models. Unlike vanilla RNNs, Bayesian RNNs are stochastic and it is possible to compute average and variance for a prediction. Using a prior to integrate out the parameters during training also leads to a regularization effect. However, there are two major and often debated challenges of BNNs: the right choice of prior and the efficient approximation of the posterior.

With this paper, we side-step these challenges and model the uncertainty of an RNN through probabilistic state transitions between a finite number of $k$ learnable states $\mathbf{s_1}, ..., \mathbf{s}_k$. Given a state $\mathbf{h}_t$, we compute a probability distribution over the learnable states. Hence, for the same state and input, the RNN might move to different states in different runs. Instead of integrating over possible weights, as in the case of BNNs, we sum over all possible state sequences and weigh the classification probabilities by the probabilities of these sequences. Figure 2 illustrates the proposed approach and contrasts it with vanilla and Bayesian RNNs. The proposed method combines two building blocks. The first is state-regularization (Wang & Niepert, 2019) as a way to compute a probability distribution over a finite set of states in an RNN. State-regularization, however, is deterministic and therefore we utilize the second building block, the Gumbel softmax trick (Gumbel, 1954; Maddison et al., 2017; Jang et al., 2017) to sample from a categorical distribution. Combining the two blocks allows us to create

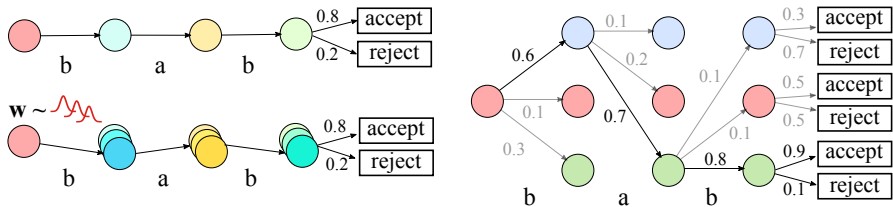

Figure 2: Illustration of several ways to model uncertainty in recurrent models. Left, top: In standard recurrent models the sequence of hidden states is identical for any two runs on the same input. Uncertainty is typically modeled with a softmax distribution over the classes (here: accept and reject). Left, bottom: Bayesian NNs make the assumption that the weights are drawn from a distribution. Uncertainty is estimated through model averaging. Right: The proposed class of RNNs assumes that there is a *finite* number of states between which the RNN transitions. Uncertainty for an input sequence is modeled through a probability distribution over possible state-transition paths.

a stochastic state RNN which can model uncertainty. Before we formulate our method, we first introduce the necessary two building blocks.

**Deterministic State-Regularized RNNs.** State regularization (Wang & Niepert, 2019) extends RNNs by dividing the computation of the hidden state $\mathbf{h}_t$ into two components. The first component is an intermediate vector $\mathbf{u}_t \in \mathbb{R}^d$ computed in the same manner as the standard recurrent component, $\mathbf{u}_t = f(\mathbf{h}_{t-1}, \mathbf{c}_{t-1}, \mathbf{x}_t)$. The second component models probabilistic state transitions between a finite set of $k$ learnable states $\mathbf{s}_1, \ldots, \mathbf{s}_k$, where $\mathbf{s}_i \in \mathbb{R}^d, i \in [1, k]$ and which can also be written as a matrix $\mathbf{S}_t \in \mathbb{R}^{d \times k}$. $\mathbf{S}_t$ is randomly initialized and learnt during backpropagation like any other network weight. At time step $t$, given an $\mathbf{u}_t$, the transition over next possible states is computed by: $\boldsymbol{\theta}_t = \varphi(\mathbf{S}_t, \mathbf{u}_t)$, where $\boldsymbol{\theta}_t = \{\theta_{t,1}, ..., \theta_{t,k}\}$ and $\varphi$ is some pre-defined function. In Wang & Niepert (2019), $\varphi$ was a matrix-vector product followed by a SOFTMAX function that ensures $\sum_{i=1}^{k} \theta_{t,i} = 1$. The hidden state $\mathbf{h}_t$ is then computed by

$$\mathbf{h}_t = g(\boldsymbol{\theta}_t) \cdot \mathbf{S}_t^\top, \quad \mathbf{h}_t \in \mathbb{R}^d, \tag{1}$$

where $g(\cdot)$ is another function, e.g. to compute the average. Because Equation (1) is deterministic, it cannot capture and estimate epistemic uncertainty.

**Monte-Carlo Estimator with Gumbel Trick.** The Gumbel softmax trick is an instance of a pathwise Monte-Carlo gradient estimator (Gumbel, 1954; Maddison et al., 2017; Jang et al., 2017). With the Gumbel trick, it is possible to draw samples $z$ from a categorical distribution given by paramaters $\boldsymbol{\theta}$, that is, $\boldsymbol{z} = \text{ONE\_HOT}(\arg\max_i[\gamma_i + \log\theta_i]), i \in [1 \dots k]$, where $k$ is the number of categories and $\gamma_i$ are i.i.d. samples from the GUMBEL$(0, 1)$, that is, $\gamma = -\log(-\log(u)), u \sim \text{UNIFORM}(0, 1)$. Because the $\arg\max$ operator breaks end-to-end differentiability, the categorical distribution $\boldsymbol{z}$ can be approximated using the differentiable softmax function (Jang et al., 2017; Maddison et al., 2017). This enables us to draw a $k$-dimensional sample vector $\boldsymbol{\alpha} \in \Delta^{k-1}$, where $\Delta^{k-1}$ is the $(k-1)$-dimensional probability simplex.

## 2.2 STOCHASTIC FINITE-STATE RNNS (ST-$\tau$)

Our goal is to make state transitions stochastic and uitilize them to measure uncertainty: given an input sequence, the uncertainty is modeled via the probability distribution over possible state-transition paths (see right half of Figure 2). We can achieve this by setting $\varphi$ to be a matrix-vector product and using $\boldsymbol{\theta}_t$ to sample from a Gumbel softmax distribution with temperature parameter $\tau$. Applying Monte Carlo estimation, at each time step $t$, we sample a distribution over state transition probabilities $\boldsymbol{\alpha}_t$ from the Gumbel softmax distribution with current parameter $\tau$, where each state transition has the probability

$$\alpha_{t,i} = \frac{\exp((\log(\theta_{t,i}) + \gamma_i)/\tau)}{\sum_{j=1}^{k} \exp((\log(\theta_{t,j}) + \gamma_j)/\tau)}, i \in [1 \dots k]. \tag{2}$$

The resulting $\boldsymbol{\alpha}_t = \{\alpha_{t,1}, ..., \alpha_{t,k}\}$ can be seen as a probability distribution that judges how important each learnable state $\mathbf{s}_t$ is. The new hidden state $\mathbf{h}_t$ can now be formed either as

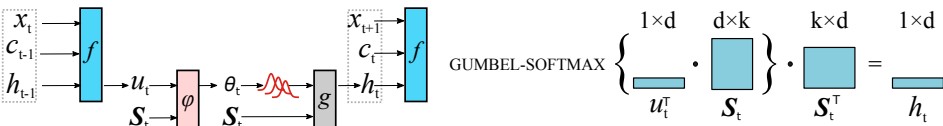

Figure 3: One step of ST-$\tau$ (for batch size $b = 1$). First, previous hidden state $\mathbf{h}_{t-1}$, previous cell state $\mathbf{c}_{t-1}$ (if given) and input $\mathbf{x}_t$, are passed into an RNN cell (e.g. LSTM). The RNN cell returns an updated hidden state $\mathbf{u}_t$ and cell state $\mathbf{c}_t$. Second, $\mathbf{u}_t$ is further processed by using the learnable states matrix $\mathbf{S}_t$ in the state transition step (Right) and returns a new hidden state $\boldsymbol{h}_t$.

an average, $\mathbf{h}_t = \sum_{i=1}^{k} \alpha_{t,i} \mathbf{s}_i$ (the "soft" Gumbel estimator), or as a one-hot vector, $\mathbf{h}_t = \text{ONE\_HOT}\big(\arg\max_i [\log(\theta_{t,i}) + \gamma_i]\big)$. For the latter, gradients can be estimated using the straight-through estimator. Empirically, we found the average to work better. By sampling from the Gumbel softmax distribution at each time step, the model is stochastic and it is possible to measure variance across predictions and, therefore, to estimate epistemic uncertainty. For more theoretical details we refer the reader to Appendix 2.3.

The parameter $\tau$ of the Gumbel softmax distribution is learned during training (Jang et al., 2017). This allows us to directly adapt probabilistic RNNs to the inherent uncertainty of the data. Intuitively, the parameter $\tau$ influences the concentration of the categorical distribution, that is, the larger $\tau$ the more uniform the distribution. Since we influence the state transition uncertainty with the learned temperature $\tau$, we refer to our model as ST-$\tau$. We provide an ablation experiment of learning $\tau$ versus keeping it fixed in Appendix E.

Figure 3 illustrates the proposed model. Given the previous hidden state of an RNN, first an intermediate hidden state $\mathbf{u}_t$ is computed using a standard RNN cell. Next, the intermediate representation $\mathbf{u}_t$ is multiplied with $k$ learnable states arranged as a matrix $\mathbf{S}_t$, resulting in $\boldsymbol{\theta}_t$. Based on $\boldsymbol{\theta}_t$, samples are drawn from a Gumbel softmax distribution with learnable temperature parameter $\tau$. The sampled probability distribution represents the certainty the model has in moving to the other states. Running the model on the same input several times (drawing Monte-Carlo samples) allows us to estimate the uncertainty of the ST-$\tau$ model.

## 2.3 Aleatoric and Epistemic Uncertainty

Let $\mathcal{Y}$ be a set of class labels and $\mathcal{D}$ be a set of training samples. For a classification problem and a given ST-$\tau$ model with states $\{\mathbf{s}_1, ..., \mathbf{s}_k\}$, we can write for every $y \in \mathcal{Y}$

$$p(y \mid \mathbf{x} = \langle \mathbf{x}_1, ..., \mathbf{x}_n \rangle) = \sum_{\langle \mathbf{h}_1, ..., \mathbf{h}_n \rangle \in \Psi} p(\langle \mathbf{h}_1, ..., \mathbf{h}_n \rangle \mid \mathbf{x}) \, q(y \mid \mathbf{h}_n) \quad (3)$$

where $\mathbf{h}_i \in \{\mathbf{s}_1, ..., \mathbf{s}_k\}$ and the sum is over all possible paths (state sequences) $\Psi$ of length $n$. Moreover, $p(\langle \mathbf{h}_1, ..., \mathbf{h}_n \rangle \mid \mathbf{x})$ is the probability of path $\psi = \langle \mathbf{h}_1, ..., \mathbf{h}_n \rangle \in \Psi$ given input sequence $\mathbf{x}$ and $q(y \mid \mathbf{h}_n)$ is the probability of class $y$ given that we are in state $\mathbf{h}_n$. Instead of integrating over possible weights, as in the case of BNNs, with ST-$\tau$ we integrate (sum) over all possible paths and weigh the class probabilities by the path probabilities. The above model implicitly defines a probabilistic ensemble of several deterministic models, each represented by a particular path. As mentioned in a recent paper about aleatoric and epistemic uncertainty in ML (Hüllermeier & Waegeman, 2019): "the variance of the predictions produced by an ensemble is a good indicator of the (epistemic) uncertainty in a prediction."

Let us now make this intuition more concrete using recently proposed measures of aleatoric and epistemic uncertainty (Depeweg et al., 2018); further discussed in (Hüllermeier & Waegeman, 2019). In Equation (19) of (Hüllermeier & Waegeman, 2019) the *total* uncertainty is defined as the entropy of the predictive posterior distribution

$$H\left[p(y \mid \mathbf{x})\right] = -\sum_{y \in \mathcal{Y}} p(y \mid \mathbf{x}) \log_2 p(y|\mathbf{x}).$$

The above term includes both aleatoric *and* epistemic uncertainty (Depeweg et al., 2018; Hüllermeier & Waegeman, 2019). Now, in the context of Bayesian NNs, where we have a distribution over the weights of a neural network, the expectation of the entropies wrt said distribution

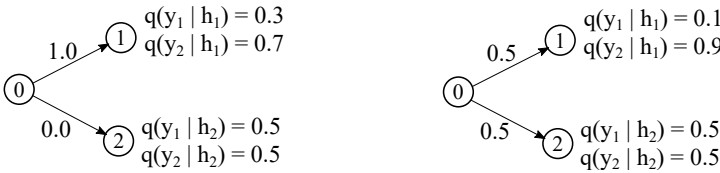

Figure 4: Two different probabilistic finite-state RNNs.

is the aleatoric uncertainty (Equation 20 in Hüllermeier & Waegeman (2019)):

$$\mathbf{E}_{p(\mathbf{w}|\mathcal{D})} H\left[p(y \mid \mathbf{w}, \mathbf{x})\right] = -\int p(\mathbf{w} \mid \mathcal{D}) \left(\sum_{y \in \mathcal{Y}} p(y \mid \mathbf{w}, \mathbf{x}) \log_2 p(y \mid \mathbf{w}, \mathbf{x})\right) d\mathbf{w}.$$

Fixing the parameter weights to particular values eliminates the epistemic uncertainty. Finally, the epistemic uncertainty is obtained as the difference of the total and aleatoric uncertainty (Equation 21 in Hüllermeier & Waegeman (2019)):

$$u_e(\mathbf{x}) := H\left[p(y \mid \mathbf{x})\right] - \mathbf{E}_{p(\mathbf{w}|\mathcal{D})} H\left[p(y \mid \mathbf{w}, \mathbf{x})\right].$$

Now, let us return to finite-state probabilistic RNNs. Here, the aleatoric uncertainty is the expectation of the entropies with respect to the distribution over the possible paths $\Psi$:

$$\mathbf{E}_{p(\psi|\mathbf{x})} H\left[p(y \mid \psi, \mathbf{x})\right] = -\sum_{\psi \in \Psi} p(\psi \mid \mathbf{x}) \left(\sum_{y \in \mathcal{Y}} p(y \mid \psi, \mathbf{x}) \log_2 p(y \mid \psi, \mathbf{x})\right),$$

where $p(y \mid \psi, \mathbf{x})$ is the probability of class $y$ conditioned on $\mathbf{x}$ and a particular path $\psi$. The epistemic uncertainty for probabilistic finite-state RNNs can then be computed by

$$u_e(\mathbf{x}) := H\left[p(y \mid \mathbf{x})\right] - \mathbf{E}_{p(\psi|\mathbf{x})} H\left[p(y \mid \psi, \mathbf{x})\right].$$

Probabilistic finite-state RNNs capture epistemic uncertainty when the equation above is non-zero. As an example let us take a look at the two ST-$\tau$ models given in Figure 4. Here, we have for input $\mathbf{x}$ two class labels $y_1$ and $y_2$, three states ($\mathbf{s}_i, i \in \{0, 1, 2\}$), and two paths. In both cases, we have that $H\left[p(y \mid \mathbf{x})\right] = -(0.3 \log_2 0.3 + 0.7 \log_2 0.7) \approx 0.8813$. Looking at the term for the aleatoric uncertainty, for the ST-$\tau$ depicted on the left side we have $\mathbf{E}_{p(\psi|\mathbf{x})} H\left[p(y \mid \psi, \mathbf{x})\right] = -(0.3 \log_2 0.3 + 0.7 \log_2 0.7) \approx 0.8813$. In contrast, for the ST-$\tau$ depicted on the right side we have $\mathbf{E}_{p(\psi|\mathbf{x})} H\left[p(y \mid \psi, \mathbf{x})\right] \approx 0.7345$. Consequently, the left side ST-$\tau$ has an epistemic uncertainty of $u_e(\mathbf{x}) = 0$ but the right side ST-$\tau$ exhibits an epistemic uncertainty of $u_e(\mathbf{x}) = 0.1468$.

This example illustrates three main observations. First, we can represent epistemic uncertainty through distributions over possible paths. Second, the more spiky the transition distributions, the more deterministic the behavior of the ST-$\tau$ model, and the more confident it becomes with its prediction by shrinking the reducible source of uncertainty (epistemic uncertainty). Third, both models are equally calibrated as their predictive probabilities are, in expectation, identical for the same inputs. Hence, ST-$\tau$ is not merely calibrating predictive probabilities but also captures epistemic uncertainty. Finally, we want to stress the connection between the parameter $\tau$ (the temperature) and the degree of (epistemic) uncertainty of the model. For small $\tau$ the ST-$\tau$ model behavior is more deterministic and, therefore, has a lower degree of epistemic uncertainty. For instance, the ST-$\tau$ on the left in Figure 4 has 0 epistemic uncertainty because all transition probabilities are deterministic. Empirically, we find that the temperature and, therefore, the epistemic uncertainty often reduces during training, leaving the irreducible uncertainty (aleatoric) to be the main source of uncertainty.

## 3 RELATED WORK

**Uncertainty**. Uncertainty quantification for safety-critical applications (Krzywinski & Altman, 2013) has been explored for deep neural nets in the context of Bayesian learning (Blundell et al., 2015; Gal & Ghahramani, 2016b; Kendall & Gal, 2017; Kwon et al., 2018). Bayes by Backprop (BBB) (Blundell et al., 2015) is a variational inference scheme for learning a distribution

over weights $\mathbf{w}$ in neural networks and assumes that the weights are distributed normally, that is, $w_i \sim \mathcal{N}(\mu, \sigma^2)$. The principles of bayesian neural networks (BNNs) have been applied to RNNs and shown to result in superior performance compared to vanilla RNNs in natural language processing (NLP) tasks (Fortunato et al., 2017). However, BNNs come with a high computational cost because we need to learn $\mu$ and $\sigma$ for each weight in the network, effectively doubling the number of parameters. Furthermore, the prior might not be optimal and approximate inference could lead inaccurate estimates (Kuleshov et al., 2018). Dropout (Hinton et al., 2012; Srivastava et al., 2014) can be seen as a variational approximation of a Gaussian Process (Gal & Ghahramani, 2016b;a). By leaving dropout activated at prediction time, it can be used to measure uncertainty. However, the dropout probability needs to be tuned which leads to a trade-off between predictive error and calibration error (see Figure 10 in the Appendix for an empirical example). Deep ensembles (Lakshminarayanan et al., 2017) offer a non-Bayesian approach to measure uncertainty by training multiple separate networks and ensembling them. Similar to BNNs, however, deep ensembles require more resources as several different RNNs need to be trained. We show that ST-$\tau$ is competitive to deep ensembles without the resource overhead. Recent work (Hwang et al., 2020) describes a sample-free uncertainty estimation for Gated Recurrent Units (SP-GRU) (Chung et al., 2014), which estimates uncertainty by performing forward propagation in a series of deterministic linear and nonlinear transformations with exponential family distributions. ST-$\tau$ estimates uncertainties through the stochastic transitions between two consecutive recurrent states.

**Calibration**. Platt scaling (Platt et al., 1999) is a calibration method for binary classification settings and has been extend to multi-class problems (Zadrozny & Elkan, 2002) and the structured prediction settings (Kuleshov & Liang, 2015). (Guo et al., 2017) extended the method to calibrate modern deep neural networks, particularly networks with a large number of layers. In their setup, a temperature parameter for the final softmax layer is adjusted only after training. In contrast, our method learns the temperature and, therefore, the two processes are not decoupled. In some tasks such as time-series prediction or RL it is crucial to calibrate during training and not post-hoc.

**Deterministic & Probabilistic Automata Extraction**. Deterministic Finite Automata (DFA) have been used to make the behavior of RNNs more transparent. DFAs can be extracted from RNNs after an RNN is trained by applying clustering algorithms like $k$-means to the extracted hidden states (Wang et al., 2018) or by applying the exact learning algorithm L* (Weiss et al., 2018). Post-hoc extraction, however, might not recover faithful DFAs. Instead, (Wang & Niepert, 2019) proposed state-regularized RNNs where the finite set of states is learned alongside the RNN by using probabilistic state transitions. Building on this, we use the Gumbel softmax trick to model stochastic state transitions, allowing us to learn probabilistic automata (PAs) (Rabin, 1963) from data.

**Hidden Markov Models (HMMs) & State-Space Models (SSMs) & RNNs**. HMMs are transducer-style probabilistic automata, simpler and more transparent models than RNNs. (Bridle, 1990) explored how a HMM can be interpreted as an RNNs by using full likelihood scoring for each word model. (Krakovna & Doshi-Velez, 2016) studied various combinations of HMMs and RNNs to increase the interpretability of RNNs. There have also been ideas on incorporating RNNs to HMMs to capture complex dynamics (Dai et al., 2017; Doerr et al., 2018). Another relative group of work is SSMs, e.g., rSLDS (Linderman et al., 2017), Kalman VAE (Fraccaro et al., 2017), PlaNet (Hafner et al., 2019) and RKN (Becker et al., 2019). They can be extended and viewed as another way to inject stochasticity to RNN-based architectures. In contrast, ST-$\tau$ models stochastic finite-state transition mechanisms end-to-end in conjunction with modern gradient estimators to directly quantify and calibrate uncertainty and the underlying probabilistic system. This enables ST-$\tau$ to approximate and extract the probabilistic dynamics in RNNs.

## 4 EXPERIMENTS

The experiments are grouped into five categories. First, we show that it is possible to use ST-$\tau$ to learn deterministic and probabilistic automata from language data (Sec. 4.1). This demonstrates that ST-$\tau$ can capture and recover the stochastic behavior of both deterministic and stochastic languages. Second, we demonstrate on classification tasks (Sec. 4.2) that ST-$\tau$ performs better than or similar to existing baselines both in terms of predictive quality and model calibration. Third, we compare ST-$\tau$ with existing baselines using out-of-distribution detection tasks (Sec. 4.3). Fourth, we conduct reinforcement learning experiments where we show that the learned parameter $\tau$ can

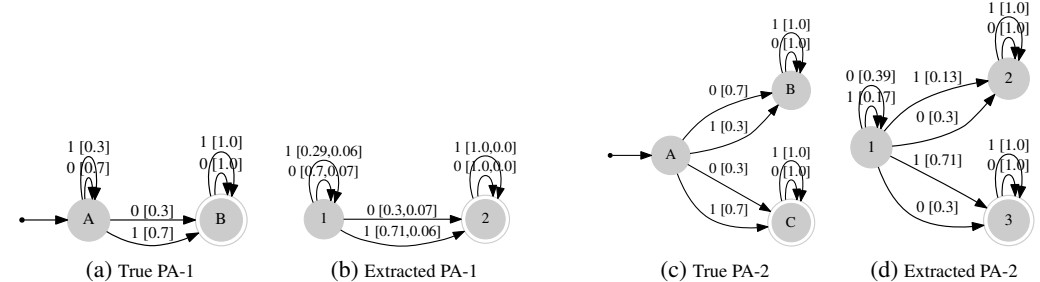

(a) True PA-1       (b) Extracted PA-1       (c) True PA-2       (d) Extracted PA-2

Figure 5: (a)+(c): ground truth probabilistic automata (PAs) used for generating training samples. (b)+(d) the PAs extracted from ST-$\tau$. The single and double circles represent *reject* and *accept* states respectively. The PA (d) resulted from a minimization of the actual extracted PA.

calibrate the exploration-exploitation trade-off during learning (Appendix D), leading to a lower sample complexity. Fifth, we report results on a regression task (Appendix C).

**Models.** We compare the proposed ST-$\tau$ method to four existing models. First, a vanilla LSTM (**LSTM**). Second, a Bayesian RNN (**BBB**) (Blundell et al., 2015; Fortunato et al., 2017), where each network weight $w_i$ is sampled from a Gaussian distribution $\mathcal{N}(\mu, \sigma^2)$, $w_i = \mu + \sigma \cdot \epsilon$, with $\sigma = \log(1 + \exp(\rho))$ and $\epsilon \sim \mathcal{N}(0, 1)$. To estimate model uncertainty, we keep the same sampling strategy as employed during training (rather than using $w_i = \mu$). Third, a RNN that employs Variational Dropout (**VD**) (Gal & Ghahramani, 2016a). In variational dropout, the same dropout mask is applied at all time steps for one input. To use this method to measure uncertainty, we keep the same dropout probability at prediction time (Gal & Ghahramani, 2016a). Fourth, a deep ensemble of a LSTM base model (Lakshminarayanan et al., 2017). For ST-$\tau$ we compute the new hidden state using the soft version of the Gumbel softmax estimator (Jang et al., 2017). All models contain only a single LSTM layer, are implemented in Tensorflow (Abadi et al., 2015), and use the ADAM (Kingma & Ba, 2015) optimizer with initial learning rate 0.001.

### 4.1 DETERMINISTIC & PROBABILISTIC AUTOMATA EXTRACTION

We aim to investigate the extent to which ST-$\tau$ can learn deterministic and probabilistic automata from sequences generated by regular and stochastic languages. Since the underlying languages are known and the data is generated using these languages, we can exactly assess whether ST-$\tau$ can recover these languages. We refer the reader to the appendix A for a definition of deterministic finite automata (DFA) and probabilistic automata (PA). The set of languages recognized by DFAs and PAs are referred to as, respectively, *regular* and *stochastic* languages. For the extraction experiments we use the GRU cell as it does not have a cell state. This allows us to read the Markovian transition probabilities for each state-input symbol pair directly from the trained ST-$\tau$.

**Regular Languages**. We conduct experiments on the regular language defined by Tomita grammar 3. This language consists of any string without an odd number of consecutive 0's after an odd number of consecutive 1's (Tomita, 1982). Initializing $\tau = 1$, we train ST-$\tau$ (with $k = 10$ states) to learn a ST-$\tau$ model to represent this grammar and then extract DFAs (see Appendix A.4 for the extraction algorithm). In principle, $k \geq$ # of classes, the model learns to select a finite set of (meaningful) states to represent a language, as shown in Appendix Figure 8. ST-$\tau$ is able to learn that the underlying language is deterministic, learning the temperature parameter $\tau$ accordingly and the extraction produces the correct underlying DFA. The details are discussed in the appendix.

**Stochastic Languages** We explore whether it is possible to recover a PA from a ST-$\tau$ model trained on the data generated by a given PA. While probabilistic deterministic finite automata (PDFAs) have been extracted previously (Weiss et al., 2019), to the best of our knowledge, this is the first work to directly learn a PA, which is more expressive than a PDFA (Denis & Esposito, 2004), by extraction from a trained RNN. We generate training data for two stochastic languages defined by the PAs shown in Figure 5 (a) and (c). Using this data, we train a ST-$\tau$ with a GRU with $k = 4$ states and we directly use the Gumbel softmax distribution to approximate the probability transitions of the underlying PA (see Appendix A.5 for more details and the extraction algorithm). Figures 5 (b, d) depict the extracted PAs. For both stochastic languages the extracted PAs indicate that ST-$\tau$ is able to learn the probabilistic dynamics of the ground-truth PAs.

| Dataset | BIH | | | IMDB | | |
|---|---|---|---|---|---|---|
| Metrics | PE | ECE | MCE | PE | ECE | MCE |
| LSTM | **1.40** | 0.78 | 35.51 | **10.42** | 3.64 | 11.24 |
| Ensembles | $\underline{1.51}\pm1e^{-3}$ | 0.72±0.10 | 31.49±8.52 | $\underline{10.56}\pm3e^{-3}$ | 3.45±1.47 | 12.16±5.68 |
| BBB | $4.69\pm2e^{-3}$ | 0.54±0.11 | **12.44**±7.65 | $10.84\pm2e^{-4}$ | $\underline{2.10}$±0.02 | $\underline{6.15}$±0.25 |
| VD | $\underline{1.51}\pm3e^{-4}$ | 0.80±0.03 | 24.71±16.70 | $\underline{10.56}\pm6e^{-4}$ | 3.41±0.07 | 14.08±0.87 |
| ST-$\tau$ $k = 5/2$ | $2.12\pm5e^{-4}$ | $\underline{0.45}$±0.03 | 23.11±12.76 | $10.95\pm5e^{-4}$ | **0.89**±0.05 | **3.70**±0.63 |
| ST-$\tau$ $k = 10$ | $2.11\pm5e^{-4}$ | **0.40**±0.05 | $\underline{21.73}$±16.15 | $11.16\pm7e^{-4}$ | 3.38±0.05 | 9.09±0.74 |

Table 1: Predictive Error (PE) and calibration errors (ECE, MCE) for the datasets BIH and IMDB (lower is better for all metrics). ST-$\tau$ offers the best and reliable trade-off between predictive error and calibration errors. Furthermore, it does not require more parameters as BBB (double) or Deep Ensemble (order of magnitude more) nor does a hyperparameter has to be tuned as in VD. Stochastic predictions are averaged across 10 independent runs and their variance is reported. Best and second best results are marked in bold and underlined (PE: bold models are significantly different at level $p \leq 0.005$). An ablation experiment with post-hoc temperature scaling is in Appendix B.3.1.

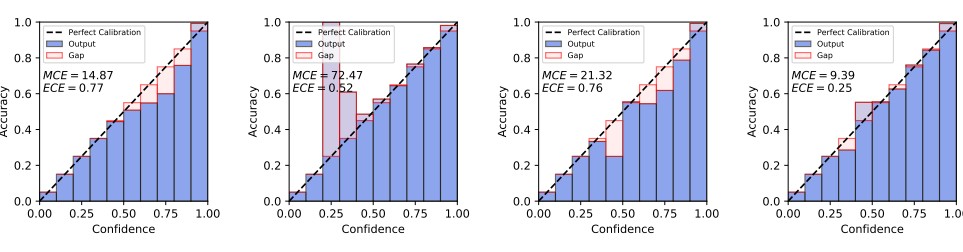

(a) LSTM (error 1.32%)   (b) BBB (error 4.29 %)   (c) VD 0.05 (error 1.45%)   (d) ST-$\tau$ (error 2.09%)

Figure 6: Calibration plots (for one run) on the BIH dataset (corresponding to the left half of Table 1) using $N = 10$ bins. ST-$\tau$ ($k = 2$) is closest to the diagonal line, that is, perfect calibration.

## 4.2 MODEL CALIBRATION

We evaluate ST-$\tau$'s prediction and calibration quality on two classification tasks. The first task is heartbeat classification with 5 classes where we use the MIT-BIH arrhythmia dataset (Goldberger et al., 2000; Moody & Mark, 2001). It consists of 48 half-hour excerpts of electrocardiogram (ECG) recordings. To preprocess the recordings we follow Kachuee et al. (2018) (Part III.A). The second task is sentiment analysis where natural language text is given as input and the problem is binary sentiment classification. We use the IMDB dataset (Maas et al., 2011), which contains reviews of movies that are to be classified as being positive or negative. For further dataset and hyperparameter details, please see Appendix B.2.

We compute the output of each model 10 times and report mean and variance. For the deep ensemble, we train 10 distinct LSTM models. For VD, we tuned the best dropout rate in $\{0.05, 0.1, 0.15\}$ and for BBB we tuned $\mu = \{0, 0.01\}$ and $\rho = \{-1, -2, -3, -4\}$, choosing the best setup by lowest predictive error achieved on validation data. For ST-$\tau$, we evaluate both, setting the number of states to the number of output classes ($k = 5/2$, BIH and IMDB, respectively) and to a fixed value $k = 10$. We initialize with $\tau = 1$ and use a dense layer before the softmax classification. For more details see Appendix, Table 4. With a perfectly calibrated model, the probability of the output equals the confidence of the model. However, many neural networks do not behave this way Guo et al. (2017). To assess the extent to which a model is calibrated, we use reliability diagrams as well as Expected Calibration Error (ECE) and Maximum Calibration Error (MCE). Please see Appendix B.1 for details. For VD, the best dropout probability on the validation set is 0.05. Lower is better for all metrics. For PE, all models are marked bold if there is no significant difference at level $p \leq 0.005$ to the best model.

**Results**. The results are summarized in Table 1. For the BIH dataset, the vanilla LSTM achieves the smallest PE with a significant difference to all other models at $p \leq 0.005$ using an approximate randomization test (Noreen, 1989). It cannot, however, measure uncertainty and suffers from higher ECE and MCE. Similarly, VD exhibits a large MCE. The situation is reversed for BBB were we find a vastly higher PE, but lower MCE. In contrast, ST-$\tau$ achieves overall good results: PE is only

| Datasets | IMDB(In)/Customer(Out) | | | IMDB(In)/Movie(Out) | | |
| Method | Accuracy | O-AUPR | O-AUROC | Accuracy | O-AUPR | O-AUROC |
| --- | --- | --- | --- | --- | --- | --- |
| LSTM (max. prob.) | 87.9 | 72.5 | 77.3 | 88.1 | 66.7 | 71.6 |
| VD 0.8 | **88.5** | 74.8 | 80.6 | 87.5 | 69.3 | 74.7 |
| BBB $\rho = -3$ | 87.6 | 67.4 | 72.0 | 87.6 | 67.1 | 71.9 |
| ST-$\tau$ $k = 10$ | 88.3 | **80.1** | **84.5** | **88.1** | **75.1** | **81.0** |
| VD 0.8 | **88.5** | 67.8 | 76.5 | 87.5 | 63.8 | 71.8 |
| BBB $\rho = -3$ | 87.6 | 76.0 | 75.4 | 87.6 | **76.8** | 75.6 |
| ST-$\tau$ $k = 100$ | 86.5 | **78.9** | **82.8** | 85.9 | 74.0 | **78.7** |
| ST-$\tau$ $k = 10$ | 88.3 | 65.0 | 76.5 | **88.1** | 64.1 | 75.1 |
| Ensembles (max.prob.) | **88.6** | 78.9 | **84.4** | **88.3** | 74.5 | 79.6 |
| Ensembles (variance) | **88.6** | 79.7 | 84.0 | **88.3** | 75.8 | 79.9 |

Table 2: Results (averaging on 10 runs for VD, BBB, ST-$\tau$. Ensembles are based on 10 models) of the out-of-distribution (OOD) detection with max-probability based (top), variance of max-probability based (middle) and ensembles (bottom). ST-$\tau$ exhibits very competitive performance.

slightly higher than the best model (LSTM) while achieving the lowest ECE and the second lowest MCE. The calibration plots of Figure 6 show that ST-$\tau$ is well-calibrated in comparison to the other models. For the IMDB dataset, ST-$\tau$ has a slightly higher PE than the best models, but has the lowest ECE and MCE offering a good trade-off. The calibration plots of IMDB can be found in the Appendix, Figure 9. In addition to achieving consistently competitive results across all metrics, ST-$\tau$ has further advantages compared to the other methods. The deep ensemble doubles the number of parameters by the number of model copies used. BBB requires the doubling of parameters and a carefully chosen prior, where ST-$\tau$ does only require a slight increase in number of parameters compared to a vanilla LSTM. VD requires the tuning of the hyperparameter for the dropout probability, which leads to a trade-off between predictive and calibration errors (see Appendix B.3.2).

## 4.3 Out-Of-Distribution Detection

We explore the ability of ST-$\tau$ to estimate uncertainty by making it detect out-of-distribution (OOD) samples following prior work (Hendrycks & Gimpel, 2017). The in-distribution dataset is IMDB and we use two OOD datasets: the Customer Review test dataset (Hu & Liu, 2004) and the Movie review test dataset (Pang et al., 2002), which consist of, respectively, 500 and 1,000 samples. As in Hendrycks & Gimpel (2017), we use the evaluation metrics AUROC and AUPR. Additionally, we report the accuracy on in-distribution samples. For VD we select the dropout rate from the values $\{0.05, 0.1, 0.2\}$ and for BBB we select the best $\mu = \{0, 0.01\}$ and $\rho = \{-1, -2, -3, -4\}$, based on best AUROC and AUPR. For ST-$\tau$ we used $c = 10$ and $c = 100$. Beside using the maximum probability of the softmax (MP) as baseline (Hendrycks & Gimpel, 2017), we also consider the variance of the maximum probability (Var-MP) across 10 runs. The number of in-domain samples is set to be the same as the number of out-of-domain samples from IMDB (Maas et al., 2011). Hence, a random baseline should achieve 50% AUROC and AUPR.

**Results**. Table 2 lists the results. ST-$\tau$ and deep ensembles are the best methods in terms of OOD detection and achieve better results for both MP and Var-MP. The MP results for ST-$\tau$ are among the best showing that the proposed method is able to estimate out-of-distribution uncertainty. We consider these results encouraging especially considering that we only tuned the number of learnable finite states $c$ in ST-$\tau$. Interestingly, a larger number of states improves the variance-based out-of-distribution detection of ST-$\tau$. In summary, ST-$\tau$ is highly competitive in the OOD task.

## 5 Discussion and Conclusion

We proposed ST-$\tau$, a novel method to model uncertainty in recurrent neural networks. ST-$\tau$ achieves competitive results relative to other strong baselines (VD, BBB, Deep Ensembles), while circumventing some of their disadvantages, e.g., extensive hyperparameters tuning and doubled number of parameters. ST-$\tau$ provides a novel mechanism to capture the uncertainty from (sequential) data over time steps. The key characteristic which distinguishes ST-$\tau$ from baseline methods is its ability to model discrete and stochastic state transitions using modern gradient estimators at each time step.

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

## A  DETERMINISTIC & PROBABILISTIC AUTOMATA EXTRACTION

For the ST-$\tau$ models we use an embedding layer and an LSTM layer (both with 100 hidden units) and a dense layer which accepts the last hidden state output and has two output neurons (accept, reject). The training objective aims to minimize a cross-entropy loss for a binary classification problem (accept or reject).

### A.1  DEFINITION OF DETERMINISTIC FINITE AUTOMATA

A Deterministic Finite Automata (DFA) is a 5-tuple $(\mathcal{Q}, \Sigma, \delta, q_0, F)$ consisting of a finite set of states $\mathcal{Q}$; a finite set of input tokens $\Sigma$ (called the input alphabet); a transition function $\delta : \mathcal{Q} \times \Sigma \rightarrow \mathcal{Q}$; a start state $q_0$; and a set of accept states $F \subseteq \mathcal{Q}$. Given a DFA and an input, it is possible to follow how an accept or reject state is reached. DFAs can be extracted from RNNs in order to offer insights into the workings of the RNN, making it more interpretable. SR-RNNs (Wang & Niepert, 2019) extract DFAs from RNNs by counting the number of transitions that have occurred between a state and its subsequent states, given a certain input (Schellhammer et al., 1998). However, this extraction method is deterministic and cannot give any uncertainty estimates for the extracted DFA. By adding stochasticity using the Gumbel softmax distribution, we can additionally offer uncertainty measures for the state transitions.

### A.2  EXTRACTING AUTOMATA FOR REGULAR LANGUAGES

With this experiment, we want to explore two features of ST-$\tau$. First, we want to understand how $\tau$ changes as training progresses (see Figure 7 (a)). At the beginning of training, $\tau$ first increases, allowing the model to explore the state transitions and select the states which will represent the corresponding grammar (in our example, the model selects 5 out of 10 states to represent Tomita 3, see Figure 7 (b, c)). Later, $\tau$ decreases and the transition uncertainty is calibrated to have tighter bounds, becoming more deterministic). Second, we want to see if ST-$\tau$ can model the uncertainty in transitions and adaptively learn to calibrate the state transition distributions. For this, we extract DFAs at two different iterations (see Figure 7 (b, c)). After 50k iterations, a correct DFA can be extracted. However, the transitions are not well calibrated. The ideal transition should be deterministic and have transition probability close to 1. For example, at state 7, for input "*1*", only 53% of the time the model transitions to the correct state 9. In contrast, after 250k iterations, the transitions are well calibrated and all transition are almost deterministic. At the same time, $\tau$ has lowered, indicating that the model has become more certain.

### A.3  DEFINITION OF PROBABILISTIC AUTOMATA

The stochasticity in ST-$\tau$ also allows us to extract Probabilistic Automata (PA) (Rabin, 1963) which have a stronger representational power than DFAs. A PA is defined as a tuple $(\mathcal{Q}, \Sigma, \delta, q_0, F)$ consisting of a finite set of states $\mathcal{Q}$; a finite set of input tokens $\Sigma$; a transition function $\delta : \mathcal{Q} \times \Sigma \rightarrow \mathcal{P}$, where $\mathcal{P}$ is the transition probability for a particular state, and $\mathcal{P}(\mathcal{Q})$ denotes the power set of $\mathcal{Q}$; a start state $q_0$ and a set of accept states $F \subseteq \mathcal{Q}$.

### A.4  EXTRACTING DFAS WITH UNCERTAINTY INFORMATION

Let $p(s_j|s_i, x_t)$, $i, j \in \{1, ..., k\}$, $x_t \in \Sigma$ be the probability of the transition to state $s_j$, given current state $s_i$ and current input symbol $x_t$ at a given time step $t$. We query each training sample to model and record the transition probability for each input symbol. The DFA extraction algorithms in (Schellhammer et al., 1998; Wang et al., 2018) are based on the count of transitions. In contrast, our extraction algorithm utilizes the transition probability, as described in Algorithm 1.

### A.5  EXTRACTING PROBABILISTIC AUTOMATA

We first define two probabilistic automata as shown in Figure 5(a)(c) to generate samples from it[2]. We generated 10,170 samples for stochastic language 1 (abbreviated SL-1) with PA-1 (Figure 5(a))

---

[2]We generate samples without combining identical samples. For instance, consider a sequence drawn from PA-2 which has probability 0.7 to be rejected and probability 0.3 to be accepted. In this case, we generate 10

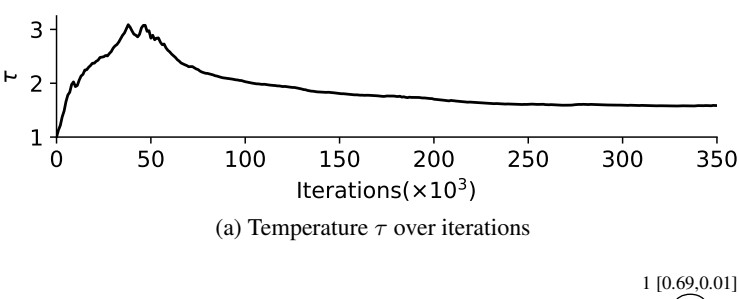

(a) Temperature $\tau$ over iterations

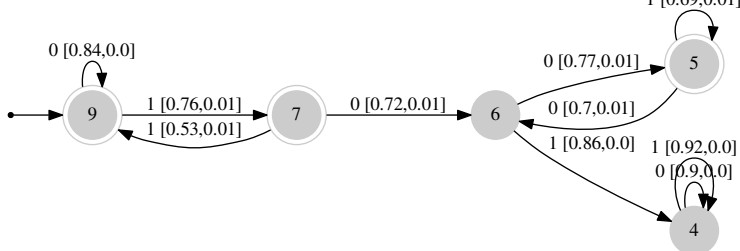

(b) Extracetd DFA after $50 \times 10^3$ iterations, $\tau = 2.98$.

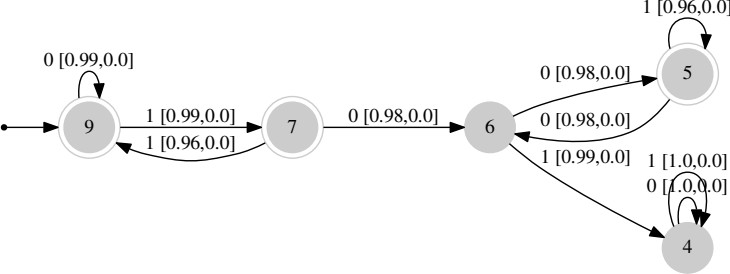

(c) Extracetd DFA after $250 \times 10^3$ iterations, $\tau = 1.61$

Figure 7: (a) As training progresses the learned temperature $\tau$ decreases. This indicates that the model has reduced its epistemic uncertainty. (b, c) The extracted DFAs at different iterations on the Tomita grammar 3 with input symbol and, in square brackets, the transition probability and uncertainty, quantified by the variance. At the earlier stage in training (b), the probabilities are still far away from being deterministic, which is also indicated by the higher value of $\tau$ and the non-zero variance. Later (c), the transition probabilities are close to being deterministic, the temperature has lowered and there is no more epistemic uncertainty. States 1-3, 8, 10 are missing as the model choose not to use these states.

and sample length $l \in [1, 9]$. For SL-2 with PA-2, we generated 20,460 samples with sample length $l \in [1, 10]$. The learning procedure is described in Algorithm 2. We use the Gumbel softmax distribution in ST-$\tau$ to approximate the probability distribution of next state $p(s_j|s_i, x_t)$. To achieve this goal, we set the number of states $k$ to an even number and force the first half of the state to "reject" state and the second half of states to be "accept". This allows us to ensure that the model models both "reject" and "accept" with the same number of states.

In the main part of the paper we report results when setting ST-$\tau$ to have $k = 4$ states for SL=2. Here, we additionally present the results for $k = 6$ in Figure 8, which yields comparable results, showing that ST-$\tau$ is not overly sensitive to the number of states. Is is however, helpful to have the same number of accept and reject states.

---

training samples with the sequence, 7 of which are labeled "0" (reject) and 3 of which are labeled "1" (accept) in expectation.

---

**Algorithm 1** Extracting DFAs with Uncertainty Information

---

**Input:** model $\mathcal{M}$, dataset **D**, alphabet $\Sigma$, start token $x_0$
**Output:** transition function $\delta$
 1: Initialize an empty dictionary $Z[s_j|s_i, x_t] = \varnothing$
 2: Compute the probability distribution over $k$ states when input $x_0$:
  $p_{1:k} = \mathcal{M}(x_0)$, $s_j = \arg\max_{i \in \{1,...,k\}}[p_{1:k}]$
 3: Set $s_i = s_j$, Update $Z[s_j|x_0] = \{p_j\}$
 4: **for** $\mathbf{x} = (x_1, x_2, ..., x_T) \in \mathbf{D}$ **do**
 5:   **for** $t \in [1, ..., T]$ **do**
 6:     $p_{1:k} = \mathcal{M}(s_i, x_t)$, $s_j = \arg\max_{i \in \{1,...,k\}}[p_{1:k}]$,
 7:     Set $s_i = s_j$, Update $Z[s_j|s_i, x_t]$:
 8:     $Z[s_j|s_i, x_t] = Z[s_j|s_i, x_t] \cup \{p_j\}$,
 9:   **end for**
10: **end for**
11: Compute transition function $\delta$, transition probability mean $p_u$ and variance $p_{var}$:
12: **for** $i, j \in \{1, ..., k\}$ and $x_t \in \Sigma$ **do**
13:   $p_u = mean(Z[(s_j|s_i, x_t)])$
14:   $p_{var} = var(Z[(s_j|s_i, x_t)])$
15:   $\delta(s_j|s_i, x_t) = \arg\max_{j \in \{1,...,k\}} p_u$
16: **end for**

---

**Algorithm 2** Extracting PAs with ST-$\tau$

---

**Input:** dataset **D**
**Output:** network loss $L$
  **for** $\mathbf{x}, \mathbf{y} \in \mathbf{D}$ **do**
    initialize $h_0$ with the $1^{st}$ state $s_1$, $s_1 \in \mathbf{S} = \{s_{1:k}\}$
    $h_0 = s_1$, $\hat{\mathbf{y}} = 0$
    **for** $x_t \in \mathbf{x}, t \in [1, ..., T]$ **do**
      $z_t = sigma(W_z x_t + U_z h_{t-1})$
      $r_t = sigma(W_r x_t + U_r h_{t-1})$
      $g_t = tahn(W_g x_t + U_g h_{t-1})$
      $u_t = z_t \odot h_{t-1} + (1 - z_t) \odot g_t$
      $p_t = \mathbf{S} u_t^\top$
      $\hat{p}_t \sim \text{GUMBEL}(p_t, \tau)$
      $h_t = \hat{p}_t^\top \mathbf{S}$
      $\hat{\mathbf{y}} = \hat{p}_t$
    **end for**
    **if** $\mathbf{y} = 0$ **then**
      $\mathbf{y} = [\underbrace{0, 0, ..0}_{k/2}, \underbrace{1, 1, ..1}_{k/2}]$
    **else**
      $\mathbf{y} = [\underbrace{1, 1, ..1}_{k/2}, \underbrace{0, 0, ..0}_{k/2}]$
    **end if**
    Compute cross-entropy loss:
    $L = \text{CROSS-ENTROPY}(\mathbf{y}, \hat{\mathbf{y}})$
  **end for**

---

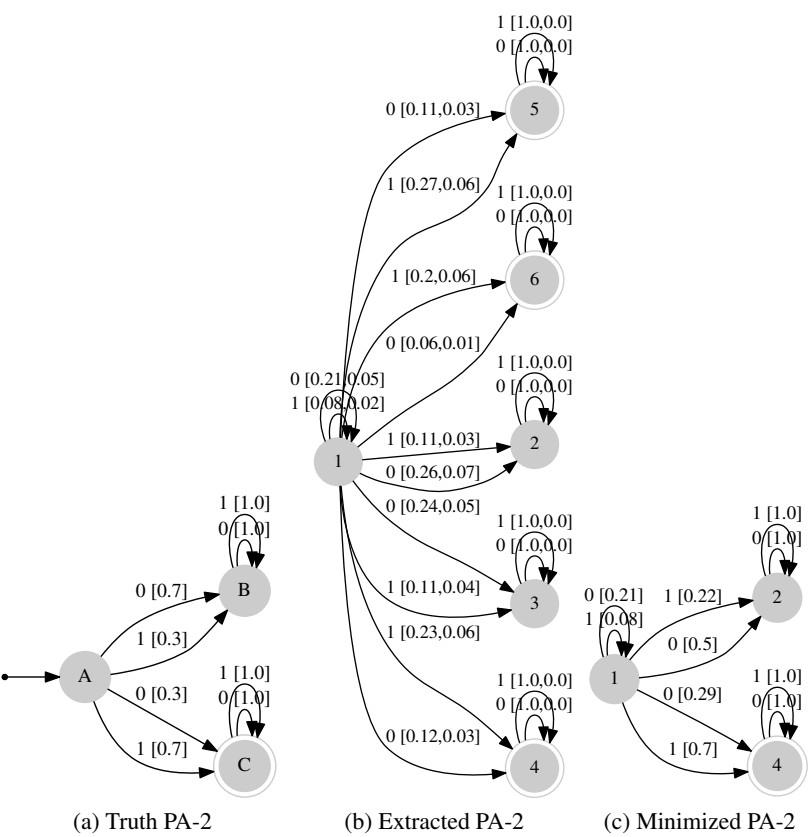

(a) Truth PA-2      (b) Extracted PA-2      (c) Minimized PA-2

Figure 8: True PA (a) used for generating training samples and the respective extracted PAs (b, c) from the trained ST-$\tau$ models with $k = 6$ states. We merged reject and accept nodes in (b) to retain one reject (start state 1 is not merged) and accept node in (c). For a given state and input symbol, the probability of accept and reject for (a) and (c) are nearly equivalent. For example, at state $s_A$ in (a), input "0", $p(accept|s_A, \text{"0"}) = 0.3 \approx 0.29 = p(accept|s_1, \text{"0"})$ and $p(accept|s_A, \text{"0"}) = 0.7 \approx 0.71 = 0.5 + 0.21 = p(accept|s_1, \text{"0"})$.

# B  MODEL CALIBRATION

We address the problem of supervised multi-class sequence classification with recurrent neural networks. We follow the definitions and evaluation metrics as in (Guo et al., 2017). Given input $X \in \mathcal{X}$ and ground-truth label $Y \in \mathcal{Y}$, a probabilistic classifier $m(X) = (\hat{Y}, \hat{P})$. The $\hat{Y} = \{\hat{y}_1, ..., \hat{y}_k\}, \hat{P} = \{\hat{p}_1, ..., \hat{p}_k\}$ present the predicted class label and confidence (the probability of correctness) over $k$ classes and $\sum_{i=1}^{k} \hat{p}_i = 1$.

***Definition of Calibration*** A model is perfectly calibrated if the confidence estimation equals the true probability, that is, $\mathbb{P}(\hat{Y} = Y | \hat{P} = p) = p, p \in [0, 1]$.

## B.1  EVALUATION OF CALIBRATION

**Reliability Diagrams** (DeGroot & Fienberg, 1983; Niculescu-Mizil & Caruana, 2005) visualise whether a model is over- or under-confident by grouping predictions into bins according to their prediction probability. The predictions are grouped into $N$ interval bins (each of of size $1/N$) and the accuracy of samples $y_i$ wrt. to the ground truth label $\hat{y}_i$ in each bin $b_n$ is computed as:

$$\text{acc}(b_n) = \frac{1}{|b_n|} \sum_{i}^{I} \mathbf{1}(\hat{y}_i = y_i), \tag{4}$$

where $i$ indexes all examples that fall into bin $b_n$. Let $\hat{p}_i$ be the probability for sample $y_i$, then average confidence is defined as

$$\text{conf}(b_n) = \frac{1}{|b_n|} \sum_{i}^{I} \hat{p}_i. \tag{5}$$

A model is perfectly calibrated if $\text{acc}(b_n) = \text{conf}(b_n), \forall n$ and in a diagram the bins would follow the identity function. Any derivation from this represents miscalibration.

Based on the accuracy and confidence measures, two calibration error metrics have been introduced (Naeini et al., 2015).

**Expected Calibration Error (ECE)**. Besides the reliability diagrams, ECE is a convenient tool to have scalar summary statistic of calibration. It computes the difference between model accuracy and confidence as a weighted average across bins,

$$\text{ECE} = \sum_{n=1}^{N} \frac{|b_n|}{m} |\text{acc}(b_n) - \text{conf}(b_n)|, \tag{6}$$

where $m$ is the total number of samples.

**Maximum Calibration Error (MCE)** is particularly important in high-risk applications where reliable confidence measures are absolutely necessary. It measures the worst-case deviation between accuracy and confidence,

$$\text{MCE} = max_{n \in \{1,...,N\}} |\text{acc}(b_n) - \text{conf}(b_n)|. \tag{7}$$

For a perfectly calibrated classifier, the ideal ECE and MCE both equal to 0.

| Dataset | IMDB | BIH |
|---|---|---|
| Train | $23k$ | $78k$ |
| Validation | $2k$ | $8k$ |
| Test | $25k$ | $21k$ |
| Max Length | 2,956 | 187 |
| # Classes | 2 | 5 |
| Type | Language | ECG |

Table 3: Overview of used datasets, if applicable, numbers are rounded down.

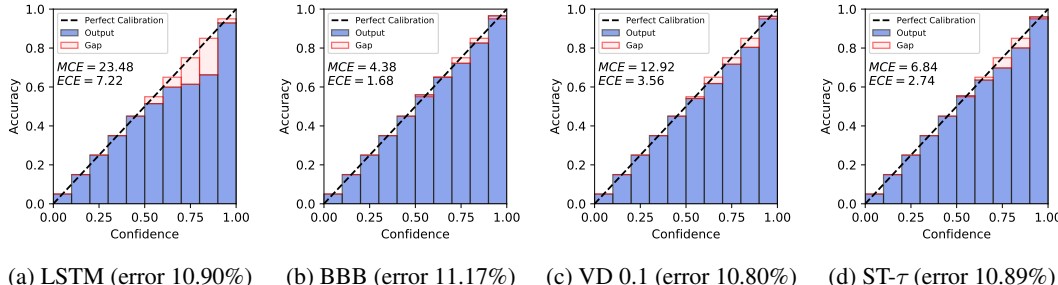

(a) LSTM (error 10.90%)  (b) BBB (error 11.17%)  (c) VD 0.1 (error 10.80%)  (d) ST-$\tau$ (error 10.89%)

Figure 9: Calibration plots for the IMDB dataset (corresponding to the right half of Table 1) using $N = 10$ bins. BBB, VD and ST-$\tau$ are all quite well calibrated on this data set. The first run is displayed. Best viewed in colour.

| Hyperparameters | IMDB | BIH |
|---|---|---|
| Hidden dim. | 256 | 128 |
| Learning rate | 0.001 | 0.001 |
| Batch size | 8 | 256 |
| Validation rate | $1k$ | $1k$ |
| Maximum validations | 20 | 50 |
| ST-$\tau$ # states | 2 | 5 |
| BBB $\mu$ | 0.0 | 0.01 |
| BBB $\rho$ | -3 | -3 |
| VD Prob. | 0.1 | 0.05 |

Table 4: Overview of the different hyperparameters for the different datasets. Validation rate indicates after how many updates validation is performed.

|  |  | PE | ECE | MCE |
|---|---|---|---|---|
| **BIH** | LSTM | **1.40** | 0.30 | 13.02 |
|  | Ensemble | $1.51 \pm 1e^{-3}$ | **0.22**±0.05 | 21.90±16.62.52 |
|  | BBB | $4.69 \pm 2e^{-3}$ | 0.36±0.09 | **10.43**±7.86 |
|  | VD 0.05 | $1.51 \pm 3e^{-4}$ | 0.27±0.02 | 23.60±18.66 |
|  | ST-$\tau$ | $2.12 \pm 5e^{-4}$ | 0.45±0.03 | 15.38±7.91 |
| **IMDB** | LSTM | **10.42** | 1.19 | 5.82 |
|  | Ensemble | $10.56 \pm 3e^{-3}$ | 1.26±0.56 | 5.87±2.40 |
|  | BBB | $10.84 \pm 2e^{-4}$ | **0.63**±0.02 | **2.69**±0.33 |
|  | VD 0.1 | $10.56 \pm 6e^{-4}$ | 2.34±0.01 | 10.86±1.33 |
|  | ST-$\tau$ | $10.95 \pm 5e^{-4}$ | 1.00±0.04 | 3.84±0.70 |

Table 5: Same as table1 but with post-hoc temperature scaling. Among the models (Ensemble, BBB, VD) that can estimate uncertainty, ST-$\tau$ is very competitive, i.e., the second best on both the BIH and the IMDB datsaet. LSTM is not able to provide any uncertainty information. Predictive Error (PE) and calibration errors (ECE, MCE) for the various RNNs on the datasets BIH and IMDB (lower is better for all metrics). ST-$\tau$ offers the best and reliable trade-off between predictive error and calibration errors. Furthermore, it does not require double the parameters as BBB nor does a hyperparameter have to be tuned as in VD. Stochastic predictions are averaged across 10 independent runs and their variance is reported. For VD, we report the best dropout probability on the validation set. Best and the second best results are marked in bold and underlined (PE: bold models are significantly different at level $p \leq 0.005$ to non-bold models).

## B.2 DATASET AND HYPERPARAMETER DETAILS

For the experiments of Section 4.2, we provide an overview of the used datasets in Table 3 and give details on the different hyperparameters used in the experiments in Table 4. On BIH, we use the training / test split of (Kachuee et al., 2018), however we additionally split off 10% of the training

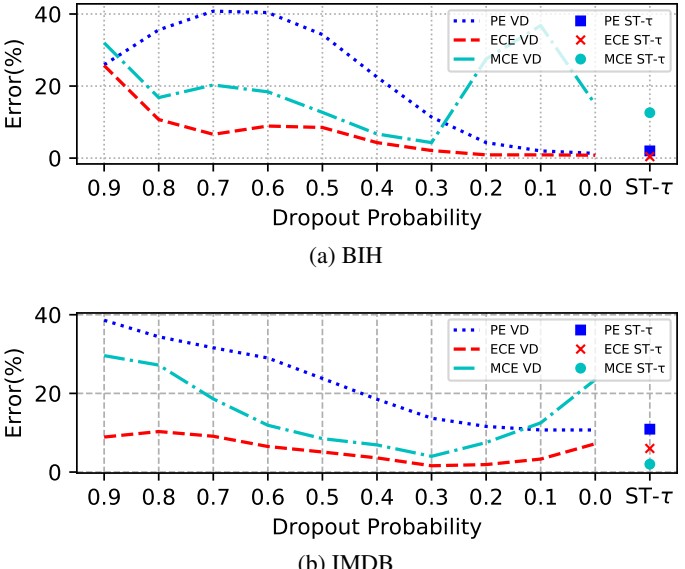

Figure 10: For VD we plot various dropout probabilities at prediction time and their corresponding error rates (PE, ECE, MCE). Additionally we report the results of ST-$\tau$. For VD, while lowering the probability rate decreases the predictive error (PE), it at the same time increases MCE and/or ECE. On the other hand, ST-$\tau$ combines the best of both worlds without having to tune any parameters.

data to use as a validation set. On IMDB, the original dataset consist of 25k training and 25k test samples, we split 2k from training set as validation dataset. The word dictionary size is limited to 5,000.

## B.3 Additional Experiments

Here we report three groups of additional or ablation experiments: (1) All baseline method and ST-$\tau$ with directly employing post-training temperature scaling. (2) The trade-off between predictive and calibrate performance with different dropout ratio in variational dropout (VD). (3) Additional calibration plots for the IMDB dataset.

### B.3.1 Experiments with Temperature Scaling

For classification calibration experiments, the post-hoc temperature scaling can also be used to calibrate models. However, please note, post-hoc temperature scaling can not be used when we need to calibrate a model during training stages, for example, the tasks like DFA or PA extraction, reinforcement learning tasks.

Table 5 reports the results with temperature scaling where temperature is tuned on valid set. Among the models (Ensemble, BBB, VD) that can estimate uncertainty, ST-$\tau$ is very competitive, for example, the second best on both the BIH and the IMDB datsaet. While LSTM can achieve better predictive performance and sometimes better calibration performance, LSTM is not able to provide any uncertainty information.

### B.3.2 Experiments with Different Dropout Rate

To gain a better understanding of how crucial the hyperparameter of VD is, we investigate the effect of the dropout probability during prediction with VD. We perform an experiment where we vary the dropout probability in the range of $[0.0, 0.9]$ with increments of $0.1$. In Figures 10 (a, b) we plot the result for BIH and IMDB, respectively, reporting PE, ECE and MCE for the various VD settings as well as ST-$\tau$ .

On both datasets, for VD, the point of lowest PE does not necessarily coincide with the points of lowest MCE and/or ECE. For example on BIH, VD achieves the lowest PE when dropout is switched

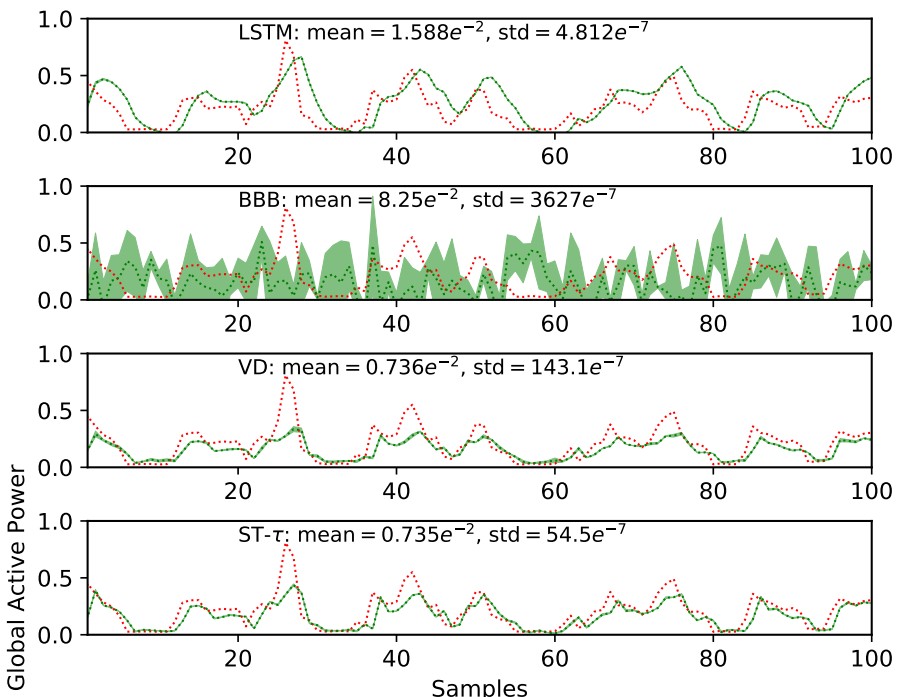

Figure 11: The test mean squared error (MSE) of different methods on predicting power consumption over time steps (x axis). The results are based on averaging 3 runs. The shaded areas present standard deviation, the first 100 samples are displayed. ST-$\tau$ and VD provide the best predictive performance, while ST-$\tau$ exhibits tighter uncertainty bounds.

off (0.0), but then uncertainty cannot be measured. On the other hand, choosing the next lowest PE results in a high MCE. In contrast, ST-$\tau$ directly achieves good results for all three metrics. Similarly, on IMDB, at the point where VD has the lowest PE is also has highest MCE. In conclusion, VD requires careful tuning which comes with choosing a trade-off between the different metrics, while ST-$\tau$ achieves good results directly, without any tuning.

### B.3.3 CALIBRATION PLOTS FOR THE IMDB DATASET

Figure 9 shows the calibration plots for IMDB. For the binary classification task BBB achieves the best calibration performance and VD achieves the best predictive performance. It should be noted that ST-$\tau$ achieves the best trade-off between predictive and calibration performance without doubling parameters (for BBB) and without tuning dropout rate for (VD).

## C REGRESSION

Calibration plays an important role in probabilistic forecasting. We consider a time-series forecasting regression task using the individual household electric power consumption dataset.[3] The goal of the task is to predict the global active power at the current time (t) given the measurement and other features of the previous time step. The dataset was sampled at the time step of an hour (the original data are given in minutes), which leads to 34,588 samples. We split it 25,000/2,000/7,588 for training/validation/test. One LSTM layer with 100 hidden units is used for the baselines (LSTM, BBB and VD) and ST-$\tau$ (the number of states is set to 10). The evaluation metric is mean squared error (MSE) and we use the model with lowest MSE on the validation dataset at test stage.

---

[3]http://archive.ics.uci.edu/ml/datasets/Individual+household+electric+power+consumption The data is preprocessed by following https://www.kaggle.com/amirrezaeian/time-series-data-analysis-using-lstm-tutorial

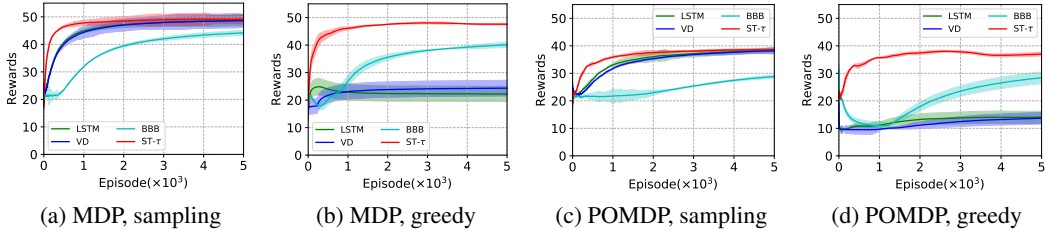

| (a) MDP, sampling | (b) MDP, greedy | (c) POMDP, sampling | (d) POMDP, greedy |

Figure 12: Average cumulative reward and standard deviation (in log scale, the shade areas) over time steps (episode) on the Cartpole task. Results are averaged over 5 randomly initialized runs. In all cases ST-$\tau$ achieves a higher averaged cumulative reward given lower sample complexity.

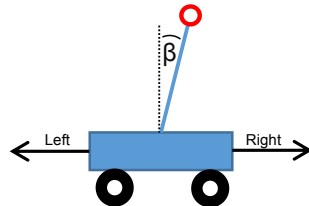

Figure 13: In the cartpole task the goal is to balance the pole upright by moving the cart left or right at each time step ($\beta$ is the angle).

Figure 11 presents the performance. LSTM, VD and ST-$\tau$ perform very well at this task, achieving MSE close to zero. BBB performs worse than the other methods. For uncertainty estimation, BBB, VD and ST-$\tau$ are able to provide uncertainty information alongside the predictive score. BBB gives high uncertainty, while VD and ST-$\tau$ are more confident in their prediction, with ST-$\tau$ offering the tightest uncertainty bounds.

## D  REINFORCEMENT LEARNING

We explore the ability of ST-$\tau$ to learn the exploration-exploitation trade-off in reinforcement learning. To demonstrate this empirically, we evaluate ST-$\tau$ in the continuous control environment CART-POLE (Figure 13) of OpenAI Gym (Brockman et al., 2016). The objective of the task is to train an agent to balance a pole for 1 second (which equals 50 time steps) given environment observations $O$. To keep the pole in balance, the agent has to move the pole either left or right, that is, the possible actions are $\boldsymbol{a} = \{\text{LEFT}, \text{RIGHT}\}$. If the chosen action keeps the pole in balance, the agent receives a reward of 1 and continues; otherwise a reward of 0 is given and the run stops.

The environment can be formulated as a reinforcement learning problem $(S, A, P, R)$ where $S$ is a set of states. Each state $s = \{x, \hat{x}, \beta, \hat{\beta}\} \in S$ consists of $x$: cart position, $\hat{x}$: cart velocity, $\beta$: angle position, and $\hat{\beta}$: angle velocity. $A$ is the set of actions, $P$ the state transition probability, and $R$ the reward. We consider two different setups. In the first setup, the CARTPOLE environment is fully observable Markov Decision Process (MDP setup) where the agent has full access to observation $O$, that is, $O = \{x, \hat{x}, \beta, \hat{\beta}\} \in S$. In the second and more difficult setup (POMDP), the environment is partially observable, where $O = \{x, \beta\} \subset S$. For the latter, the agent cannot observe the state cart velocity and angular velocity information. It has to learn to approximate them using its recurrent connections, and thus needs to retain long-term history (Bakker, 2002; Gomez & Schmidhuber, 2005; Schäfer, 2008). The various RNN-based models are trained to output a distribution over actions at each time step $t$, that is, $\pi(\boldsymbol{a}|\boldsymbol{s_t}, \boldsymbol{a_t})$, where $\pi$ is the set of policy. For selecting the next action, we consider two policies: (1) sampling: $a_{t+1} \sim \pi(\boldsymbol{a}|\boldsymbol{s_t}, \boldsymbol{a_t})$, and (2) greedy: $a_{t+1} = \arg\max \pi(\boldsymbol{a}|\boldsymbol{s_t}, \boldsymbol{a_t})$. For all baselines, we use one LSTM layer and one dense layer with softmax to return the distribution over actions. Each layer has 100 hidden units. For VD we tuned the dropout rate $\{0.05, 0.1, 0.2\}$ and for BBB $\mu = \{0, 0.01, 0.05, 0.1\}$ and $\rho = \{-1, -2, -3, -4\}$. For ST-$\tau$ we simply set the initial temperature value to $\tau = 1$, the number of possible states to the number of actions ($k = 2$), and the next action is directly selected based on the Gumbel softmax distribution over states.

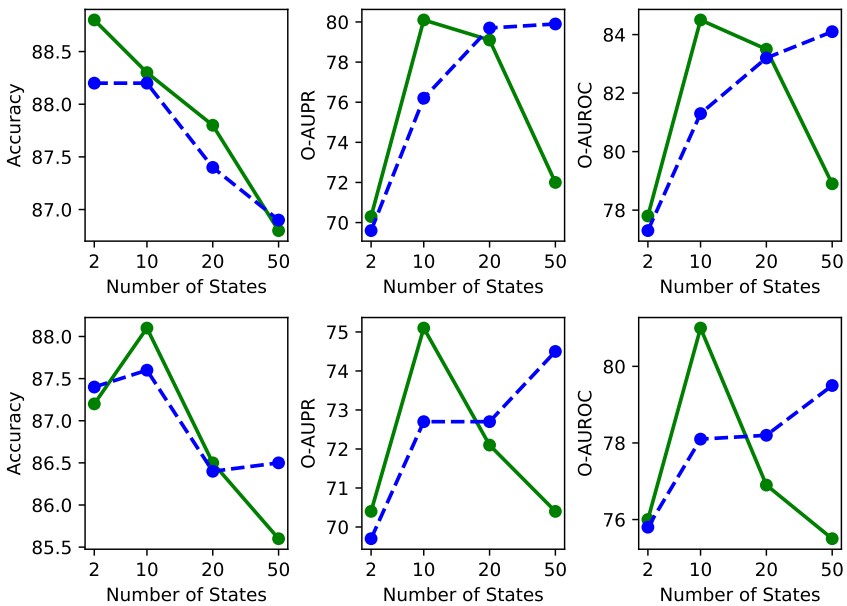

Figure 14: Apblation study for ST-$\tau$ with (1) different number of states (k=2, 10, 20, 50) and (2) learning temperature $\tau$ (green) and fixed temperature $\tau = 1$ (blue) on the Out-of-Distribution Detection (OOD) task. The max-probability based O-AUPR and O-AUROC are reported. Top: IMDB(In)/Customer(Out). Bottom: IMDB(In)/Movie(Out). The results are averaged over 10 runs. Learning the temperature $\tau$ is akin to entropy regularization (Szegedy et al., 2016; Pereyra et al., 2017; Jang et al., 2017) and adjusts the confidence (epistemic uncertainty) during training.

Results are presented in Figure 12. An important criteria for evaluating RL agents is the sample complexity (Malik et al., 2019), that is, the amount of interactions between agent and environment before a sufficiently good reward is obtained. For both environment setups and both policy types, ST-$\tau$ achieves a higher averaged cumulative reward given lower sample complexity. Moving from sampling to a greedy policy, ST-$\tau$ performance slightly drops. This is easily explained by the inherent sampling process due to the Gumbel softmax. Interestingly, it seems it is this sampling process which allows ST-$\tau$ to exhibit a lower sampling complexity in the sampling setups compared to LSTM and VD. In contrast, LSTM and VD show poor performance for the greedy policy setups because they can no longer explore by sampling from the softmax. BBB consistently performs worse than the other methods and we conjecture that this is due to the much larger number of parameters of this model, leading to a worse sampling complexity. Moving from the MDP to the POMDP, the average accumulative reward naturally drops, as the agents receive less information, but ST-$\tau$ again exhibits the best performance for both policy types.

# E    ABLATION STUDY ON OUT-OF-DISTRIBUTION DETECTION

Figure 14 depicts the results of an ablation study focusing on the number of states $k$ and whether or not the temperature $\tau$ was learned. The results are for the two IMDB Out-of-Distribution Detection (OOD) tasks from section 4.3 . The results indicate that a smaller number of states is sufficient to capture the ST-$\tau$ model's uncertainty on out-of-distribution data. Especially when the temperature parameter is learned during training (the green, solid line), ST-$\tau$ shows the best results. Increasing the number of states of ST-$\tau$ , gives the model more capacity to model uncertainty. For 50 states, fixing the temperature to a constant values works better but does not reach the accuracy of ST-$\tau$ models with fewer states and learned temperature.

