# OpenReview forum: "Uncertainty Estimation and Calibration with Finite-State Probabilistic RNNs"
_ICLR.cc/2021/Conference — ICLR 2021 Poster_

### Official Review · AnonReviewer2 · 2020-10-28
**a novel uncertainty estimation method for RNNs**

**Rating:** 7
**Confidence:** 2

**Review:**

This work proposes a novel method to estimate uncertainties in recurrent neural networks. The proposed model explicitly computes a probability distribution over a set of discrete hidden states given the current hidden state in an RNN. Leveraging the Gumbel softmax trick, the proposed method performs MC gradient estimation. A temperature parameter is also learned to control the concentration of state transition distribution. To estimate uncertainty of a given input, the proposed model is run multiple times to draw samples for estimating the mean and variance. Experiments are conducted in a variety of sequential prediction problems, including a reinforcement learning task, demonstrating the effectiveness of the proposed uncertainty estimation method.


Pros:
Estimating uncertainty of predictions is important for data-driven machine learning models, especially for detecting out-of-distribution data;
The proposed method directly quantifies and calibrates uncertainty, and therefore does not use much more parameters (compared to BNNs) and requires less parameter tuning;
The paper selects a good range of task domains and strong baseline methods, demonstrating comparable performance.

Cons:
While the proposed method demonstrates good performance on both modeling stochastic processes and estimating out-of-distribution data, it is unclear whether the method itself can separate epistemic uncertainty from aleatoric  uncertainty if both exists; meanwhile, most of the selected baseline methods focuses exclusively on estimating the epistemic uncertainty; if possible, it is desired to see a comparison of the proposed method with baseline methods that are designed to exclusively model aleatoric uncertainties for RNNs;
It is mentioned that a large number of states improves performance in the experiments for predicting OOD data; a plot for the relationship between performance and the number of states used would be useful to understand how sensitive the performance is to the number of states used;
If possible, the authors should also discuss the proposed work’s relationship with the sampling-free method of Hwang et al. [1] and how the choice of using discrete state distribution would outperform a parametric distribution.


[1] Hwang, S. J., Mehta, R. R., Kim, H. J., Johnson, S. C., & Singh, V. (2020, August). Sampling-free uncertainty estimation in gated recurrent units with applications to normative modeling in neuroimaging. In Uncertainty in Artificial Intelligence (pp. 809-819). PMLR.


------------------------------------
Update: the major concerns above have been addressed in the appendix of the updated manuscript. I'm moving my initial rating of 6 to 7.

---

> ### Author Response · Authors · 2020-11-17
> **Response to AnonReviewer2**
>
> Thank you for the detailed comments. We address the individual points you’ve raised below.
>
> Regarding measuring aleatoric and epistemic uncertainty: We have now added Appendix D that addresses formally how our proposed model class can capture uncertainty and how both aleatoric and epistemic uncertainty can be measured. Our proposed model class as well as BBB, VD, and ensemble methods can capture both types of uncertainty. In contrast, vanilla LSTMs only measure aleatoric uncertainty and serve as our aleatoric only baseline. We refer the reviewer to appendix D which also illustrates that vanilla LSTMs capture aleatoric uncertainty only.
>
> Regarding the number of states in OOD experiments: In Appendix E of the updated submission we have added ablation experiments that vary the number of states and whether $\tau$ is learnt or kept fixed for the OOD task.
>
> Comparison to Hwang et al.: Thanks for the reference, we added a discussion in the related work section. The model of Hwang et al. is deterministic whereas our proposed model is stochastic. We now explicitly discuss in Appendix D how this stochasticity allows us to capture both aleatoric and epistemic uncertainty.
>
> Thanks again for your review.

---

> > ### Comment · AnonReviewer2 · 2020-11-24
> > **detailed analysis in appendix**
> >
> > Thanks the authors for adding detailed analysis and ablation study in the Appendix.
> > All my concerns have been moderately addressed in the response.
> > I'm happy to increase my overall rating for this paper.

---

> > > ### Author Response · Authors · 2020-11-24
> > > **Thank you**
> > >
> > > Dear reviewer,
> > >
> > > We appreciate your willingness to consider the changes and additions we made and to increase your score. Your feedback has helped to improve the paper. Thank you!

---

### Official Review · AnonReviewer1 · 2020-10-29
**A good paper with extensive experimental validations and minor novelty**

**Rating:** 6
**Confidence:** 4

**Review:**

Summary:
This paper proposes a method to quantify the uncertainty for RNN. Different from  the traditional Bayesian RNN, the proposed method is more efficient. At each  time, based on the current hidden state and memory, it generates a probability  distribution over the state transition paths on the transition probability by  using the Gumbel softmax function. The next state is computed based on the weighted average of the sampled states and its uncertainty can be  qualified by the sample variance. The hyper-parameter tau of the Gumbel function  is learnt from data to better capture the inherent uncertainty in the data.

To demonstrate their method, they perform several experiments. First, they show that their model can  capture the stochastics in language better than other methods  Second, they demonstrate their  method performs better in classification on benchmark datastes than baseline methods  such as the ensemble and BBB methods in terms of both prediction accuracy and efficiency.   Third, they evaluated their method for out-of-distribution detection and their experiments again show their method performs better than the baseline methods on benchmark datasets.  Finally, they show that when applied to reinforcement  learning, their method is better than existing  methods in sample complexity.

Strengths:
The proposed method for uncertainty quantification is efficient, compared with other methods such as Bayesian RNN.  The performances of their methods have been evaluated for different tasks on benchmark datasets and show competitive performance versus the baseline methods.

Weaknesses:
First, technical novelty is minor; it is largely based on the exiting work on Gumbel function.  More importantly, is unclear why the Gumbel softmax function, even with the learnt tau parameter, can capture the data uncertainty and better theoretical justification  is needed.   Second, it is unclear how to compute the aleraeroic and epistemic uncertainties separately from their method as the latter is needed for OOD detection.  Third, it is unclear how to quantify the accuracy with the estimated uncertainty and how the improved uncertainty quantification can translate into improved performance in classification /regressions.  Fifth, the experimental comparisons are only done for baseline methods for each task.  The authors should also compare their methods to SOTA methods for each task.  Finally, they need do an ablation study on their method to figure out what contributes to their method’s improved performance for certain tasks.

---

> ### Author Response · Authors · 2020-11-17
> **Response to AnonReviewer1**
>
> Thanks for the detailed and helpful comments. We address the individual points below.
>
> Regarding novelty: The paper makes several novel contributions: (1) The stochastic component in ST-$\tau$ and its interplay with state-regularization is novel. It enables ST-$\tau$ to track the evolution of uncertainty over time steps (Figure 1); (2) The self-adaptive temperature allows the model to learn to calibrate (Section 4.2) (rather than post-hoc scaling) and to explore more efficiently in RL (Section 4.4); (3) ST-$\tau$ learns and extracts probabilistic automata directly from RNNs through novel learning and extraction algorithms (Algorithm 1 and Algorithm 2 in Appendix A). Finally, the experimental results and the thorough analysis of the proposed model class is a novel contribution.
>
> We particularly appreciate the questions regarding aleatoric and epistemic uncertainty. Motivated by said question, we have added a new Appendix D which formally shows that our proposed model class captures both aleatoric and epistemic uncertainty and how exactly this can be quantified. We would be curious to hear the reviewer’s thoughts on this update. In particular, do you think it would be helpful to integrate Appendix D into the main parts of the paper?
>
> Regarding improved performance: We quantify the uncertainty with metrics like variance, standard deviation and the ones introduced in Hendrycks & Gimpel 2016, “A Baseline for Detecting Misclassified and Out-of-Distribution Examples in Neural Networks.” Uncertainty quantification does not directly improve predictive performance, but measures the reliability of the models’ predictions.
>
> Regarding SOTA for each task:  The Bayesian NNs, variational dropout, and deep ensembles are well known SOTA methods for uncertainty estimation, which is the reason for choosing them for a consistent comparison. The objective of our work is not reaching SOTA accuracy results on each task but to explore the trade-offs between accuracy and the ability of the model to capture uncertainty.
>
> Regarding ablation: In Appendix E we have added ablation experiments that vary the number of states and whether $\tau$ is learnt or kept fixed.
>
> Thank you again for your review.

---

> > ### Comment · AnonReviewer1 · 2020-11-18
> > **Responses to authors rebuttal**
> >
> > The novelties listed in the authors response are, in my opinion, are incremental.  Moreover, my question about  why Gumbel function can capture the underlying uncertainty, in particular the epistemic uncertainty is not answered. I also disagree with the authors’ answer about measuring the  accuracy of uncertainty estimation using variance or standard deviation. Variance cannot capture the bias in  their uncertainty measurement.   Finally, without comparing to SOTA methods, it is hard to assess the effectiveness of the proposed method.

---

> > > ### Author Response · Authors · 2020-11-18
> > > **Appendix D**
> > >
> > > Dear reviewer,
> > >
> > > Thank you for your response. We appreciate that you are willing to engage with us in a discussion.
> > >
> > > Would you be so kind and take a look at Appendix D? We formally show there that our model can represent both aleatoric and epistemic uncertainty. We also show there explicitly how to quantify both uncertainties  relying on prior entropy-based definitions of aleatoric and epistemic uncertainty.
> > >
> > > The Gumbel softmax trick is a way to perform gradient estimation for a categorical distribution. As such (independent of a broader model) it cannot capture epistemic uncertainty. The way our model class captures epistemic uncertainty is through a distribution over possible paths an ST-tau model can follow. Again, we would like to point you to appendix D for more details.
> > >
> > > Could you be so kind and explain why you believe our work’s contributions are too incremental? Is there existing work that has analyzed probabilistic finite state RNNs in terms of their ability to calibrate and estimate uncertainty?
> > >
> > > We want to stress again that we do compare to SOTA methods for uncertainty estimation. Moreover, LSTMs are known to perform very well on IMDB.  Other methods for IMDB might have higher accuracy if properly tuned but are not able to quantify uncertainty. How do you suggest should we compare against SOTA method unable to estimate both/either aleatoric and epistemic uncertainty? What could sich SOTA methods be?

---

### Official Review · AnonReviewer4 · 2020-10-29
**Nice way to augment RNNs with internal randomness - but important details are missing from the paper**

**Rating:** 7
**Confidence:** 3

**Review:**

The paper proposes a novel approach for uncertainty estimation with RNNs. More precisely, the task is to both fit a model on the data and to learn the uncertainty of the fitted model at the same time.

The proposed approach fits a random model, with its randomness adjusted to the level of uncertainty. The probability of the potential outputs on a given input is then estimated by sampling the model (i.e., re-evaluating it multiple times on the same input). This, in turn, can also be used to estimate the uncertainty of the model.

One important detail that the paper does not discuss but would be important to understand is how S_t is trained/updated? (Actually, the same question goes for \tau.) In fact, referring to S_t as states is quite confusing; from the formulas it seems that they are used as weights. The authors should discuss these questions in detail.

Apart of these issues, the paper is relatively well written and the considered problem is important to various applications. The proposed model also makes sense on the high level (although the missing details make it hard to claim the same in general). Finally, empirical evaluations show the effectiveness of the method, and also that its performance is comparable - and in many cases superior - to vanilla LSTM, Bayesian RNN. RNN with variational dropout, and a deep ensemble of LSTM based model.



REMARKS

Section 2.2.
Setting \varphi to be a dot-product does not seem right: as its two attributes are \theta_t \in R^d and S_t \in S^{d x k}, the dimensions do not match. Simple matrix-vector product does work though.

In fact, Section 2 could be somewhat polished; it is not always easy to understand what is part of the proposed method, and what is explained in relation to other models only. Additionally, it would be helpful to have a brief recap at the end of the section about how the uncertainty estimation is done for the model.

In (1), t_i does not seem to be defined. Actually, should it not be {t,i}? Additionally, \alpha_i two lines below (2) should be \alpha_{t,i}, presumably.

---

> ### Author Response · Authors · 2020-11-17
> **Response to AnonReviewer4**
>
> Thank you for your helpful and thorough review. Regarding your question about the exact definition of S_t and $\tau$: S_t is indeed a weight matrix and we refer to the entries of S_t as states as they directly correspond to the states of the extracted automata in our first experiment. $\tau$ is set to an initial value (= 1) and then learned end-to-end alongside S_t and the other weights of the model. We have clarified this in the updated paper submission accordingly.
>
> Regarding the remarks: Thanks for pointing these out! We have updated the paper with the matrix-vector product and the {t,i} subscript. Additionally, we have rewritten Section 2 to be more clear, to distinguish previous work and this work better and we added a recap.
>
> We particularly appreciate the questions of you and your co-reviewers regarding the proposed model classes ability to quantify/capture (aleatoric and epistemic) uncertainty. Motivated by these remarks, we have added a new Appendix D which rigorously explains that our proposed model class captures both aleatoric and epistemic uncertainty and how exactly this can be quantified. We would be very curious to hear the reviewers’ thoughts on this update.
>
> Thanks again for your review.

---

### Official Review · AnonReviewer3 · 2020-10-31
**Review 3**

**Rating:** 7
**Confidence:** 2

**Review:**

Summary
----------

This paper presents an approach to uncertainty modeling in recurrent neural networks through a discrete hidden state. The training of this discrete model is done using a reparameterizable approximation (in particular, using the Gumbel-Softmax trick). The authors show the utility of this method on a variety of problems, including showing effective out of distribution detection and improved calibration in classification tasks.

Comments
----------

This paper presents a relatively simple idea that builds relatively directly from previous work, and uses the now common Softmax-Gumbel trick to enable differentiability. The main strength of this paper is the thorough experimental evaluation, on a wide variety of problems.

The main weakness of this paper is the very unclear presentation of the method. In section 2.1, the authors do not define all quantities, the mathematics of the method is interspersed with discussions of the approaches of others, and the writing is unclear. The authors must clarify the presentation of their method, and have this presentation be distinct from discussion of previous work.

Overall, the experimental results seem compelling and interesting. The authors should clarify their discussion of the partially observed RL task. In the partially observed task, is the agent only provided lagged measurements of the state? The presentation if quite confusing and the authors should state what this task is as clearly as possible.

Post-Rebuttal
----------
I thank the authors for their response. Both of the sections are now more clear, although the authors should make an effort to polish the narrative of the paper and the clarity of exposition throughout. The discussion of epistemic versus aleatoric uncertainty in the appendix is also interesting. I have increased my score from 6 to 7.

---

> ### Author Response · Authors · 2020-11-17
> **Response to AnonReviewer3**
>
> Thank you for your excellent review. We acknowledge that some parts of the paper lacked clarity. We have updated Section 2 and the RL task in Section 4.4 accordingly. In the partially observable task, the agent has access to the cart and angle position of the previous step, but it does not have access to the cart and angle velocity. We have clarified this in the updated submission.
>
> We have also added a formal discussion (in Appendix D) of the ways in which the proposed model class can represent (aleatoric and epistemic) uncertainty. We hope that this increases the clarity of the theoretical properties of the proposed model class further.
>
> Thanks again for your review.

---

### Author Response · Authors · 2020-11-17
**Author Response**

Dear reviewers, thank you for your thorough and helpful reviews.

We particularly appreciate the questions regarding how our proposed model measures (aleatoric and epistemic) uncertainty. Motivated by this, we have added a new Appendix D which rigorously explains that our proposed model class captures both aleatoric and epistemic uncertainty and how exactly this can be quantified. We would be very curious to hear the reviewers’ thoughts on this update.

---

### Decision · Program_Chairs · 2021-01-07
**Final Decision**

**Decision:**

Accept (Poster)

**Comment:**

This paper proposes a method to quantify the uncertainty for RNN, which is an important problem in various applications. It provides results in a variety of domains demonstrating that the proposed method outperforms baselines. However, these experiments would benefit greatly from a comparison with SOTA methods for the specific tasks in addition to the considered baselines (e.g. covariance propagation, prior network, and orthonormal certificates). The paper could also be improved by adding a theoretical justification to explain how the Gumbel softmax function is able to capture the underlying data and model uncertainty.